# Immune Activation Following Irbesartan Treatment in a Colorectal Cancer Patient: A Case Study

**DOI:** 10.3390/ijms24065869

**Published:** 2023-03-20

**Authors:** E. Titmuss, K. Milne, M. R. Jones, T. Ng, J. T. Topham, S. D. Brown, D. F. Schaeffer, S. Kalloger, D. Wilson, R. D. Corbett, L. M. Williamson, K. Mungall, A. J. Mungall, R. A. Holt, B. H. Nelson, S. J. M. Jones, J. Laskin, H. J. Lim, M. A. Marra

**Affiliations:** 1Canada’s Michael Smith Genome Sciences Centre, BC Cancer, Vancouver, BC V5Z 4S6, Canada; 2Deeley Research Centre, BC Cancer, Victoria, BC V8R 6V5, Canada; 3Department of Pathology and Laboratory Medicine, University of British Columbia, Vancouver, BC V6T 1Z7, Canada; 4Pancreas Centre BC, Vancouver, BC V5Z 1G1, Canada; 5Department of Medical Oncology, BC Cancer, Vancouver, BC V5Z 4E6, Canada; 6Department of Medical Genetics, University of British Columbia, Vancouver, BC V6T 1Z2, Canada

**Keywords:** colorectal cancer, personalised medicine, irbesartan, angiotensin receptor blocker, immunotherapy, immune response, immune activation

## Abstract

Colorectal cancers are one of the most prevalent tumour types worldwide and, despite the emergence of targeted and biologic therapies, have among the highest mortality rates. The Personalized OncoGenomics (POG) program at BC Cancer performs whole genome and transcriptome analysis (WGTA) to identify specific alterations in an individual’s cancer that may be most effectively targeted. Informed using WGTA, a patient with advanced mismatch repair-deficient colorectal cancer was treated with the antihypertensive drug irbesartan and experienced a profound and durable response. We describe the subsequent relapse of this patient and potential mechanisms of response using WGTA and multiplex immunohistochemistry (m-IHC) profiling of biopsies before and after treatment from the same metastatic site of the L3 spine. We did not observe marked differences in the genomic landscape before and after treatment. Analyses revealed an increase in immune signalling and infiltrating immune cells, particularly CD8+ T cells, in the relapsed tumour. These results indicate that the observed anti-tumour response to irbesartan may have been due to an activated immune response. Determining whether there may be other cancer contexts in which irbesartan may be similarly valuable will require additional studies.

## 1. Introduction

Colorectal cancers (CRCs) are among the most common cancers worldwide in both men and women [1] despite the potential for early detection through population-scale screening initiatives [2]. Chemotherapy, including irinotecan and oxaliplatin, and targeted therapies, such as panitumumab and bevacizumab, are standard treatments for patients with advanced disease [3]. More recently, immune checkpoint inhibitors (ICIs) have demonstrated improved clinical outcomes for subsets of patients with CRC, such as those with deficient mismatch repair (dMMR) and high tumour mutation burden (TMB) [4]. Evidence of an immune presence at the tumour site has also been associated with improved outcomes for patients on ICIs, irrespective of histology [5,6].

The Personalized OncoGenomics (POG) program at BC Cancer utilises whole genome and transcriptome analysis (WGTA) to characterize patient tumours and identify clinically actionable alterations, seeking to use these data to align patients to treatment options [7,8], some of which are off-label [9,10,11,12]. We previously described a patient with advanced, pre-treated CRC [9]. WGTA analysis identified mutations affecting mismatch repair (MMR) genes and corresponding somatic hypermutation. A variant of unknown significance affecting the angiotensin receptor *AGTR1* was detected, and there was notably high expression of the oncogenic transcription factors *FOS* and *JUN* [9,13], which are downstream targets of the angiotensin signalling pathway. Subsequently, the patient was given an angiotensin receptor blocker (ARB), irbesartan, and demonstrated an 18-month long durable, near complete radiological and metabolic response on PET/CT imaging.

This study describes WGTA and multiplex immunohistochemistry (m-IHC) of a serial biopsy collected after the patient relapsed. Our findings from this comparative analysis show that the response following irbesartan treatment correlated with an increase in immune infiltration, suggesting a treatment-associated anti-tumour immune response. These observations highlight irbesartan’s potential immune modulatory effect that could benefit patients with advanced cancer.

## 2. Results

### 2.1. Clinical Presentation

The presentation of this case was initially described in Jones et al. 2016. Following diagnosis with stage III (pT3N1) colorectal adenocarcinoma, the patient was treated with multiple cycles of combined capecitabine and oxaliplatin (Figure 1). The patient relapsed with local disease in the right psoas muscle, which was resected, and the area was treated with radiation therapy. Four years after the initial diagnosis, the tumour recurred, and the patient consented to a biopsy of the L3 spinous lesion and enrolment into the POG study (biopsy 1). At the time of enrolment, the patient exhibited extensive disease at multiple metastatic sites (Figure 1).

Following POG WGTA on the first POG biopsy, the patient started treatment with irbesartan at a daily dose of 150 mg. PET scans after five weeks revealed a systemic response (near complete functional radiological resolution) with CEA levels decreasing six-fold from 18 to 3.1 over the same time period [9]. This response continued for approximately 18 months before the patient relapsed, again in the L3 spinous process (Figure 1). At this time, the tumour was resected and submitted as a second biopsy for WGTA (biopsy 2).

### 2.2. Comparison of the Two Biopsies Reveals Few Changes in Genomic Somatic Alterations

By comparing the molecular profiles detected in biopsies 1 and 2, we sought to uncover alterations potentially linked to response and resistance to the irbesartan treatment. We compared single nucleotide variant (SNV) and copy number landscapes between the two samples and discovered they were highly similar (Figure 2A–C, Methods). In the two-year interval between the biopsies, the genome mutation burden (SNVs and indels) decreased slightly from 215.13/Mb to 191.64/Mb yet remained substantially higher than the 10 mutations/Mb often considered to be high [14,15]. Over 1600 coding SNVs and indels were shared between the two biopsies, of which 1300 were also present in a formalin-fixed paraffin-embedded (FFPE) sample taken at initial diagnosis (Figure 1). The mutations shared across all three biopsies included loss of function events in the MMR genes MLH1 (E297*), MLH3 (N674fs) and MSH3 (K381fs), indicating that the MSI and hypermutation phenotypes were early events and present at diagnosis.

In agreement with the large number of shared mutations between the biopsies, the pattern of single base substitutions (SBS), double base substitutions (DBS) and indels were almost identical in the two samples. The strongest correlations for mutation signatures were those associated with MMR, for example, SBS6 and indel signature ID2. Additionally, the copy number landscape was consistent between the two biopsies, both of which were relatively quiet. A single copy loss affecting PIK3CA was identified in both L3 biopsies, and no copy gains could be marshalled to inform clinical decision-making [9]. Taken together, the high degree of similarity between the mutation and copy number landscapes of both biopsies is consistent with the notion that the recurrence in the L3 spine was driven by the same mechanism, namely MMR.

### 2.3. Gene Expression Patterns Reveal Over-Expression of Immune Related Pathways following Treatment

Differential gene expression analysis (Methods) revealed 907 genes that were expressed at a higher level (≥2-fold) in the second biopsy and 194 genes that had lower levels of expression (≤2-fold, Figure 3A). Notably, *FOS* and *JUN* transcripts were less abundant in the second biopsy (3.3-fold and 3.4-fold, respectively). Consistent with the gene expression data, a high expression of FOS protein was detected in the first biopsy and was reduced in the second, although the difference was not statistically significant (Figure 3B). These results suggest that a blockade of the angiotensin pathway may have contributed to the reduced FOS and JUN expression in the second biopsy. Additionally, expression of the angiotensin receptor *AGTR1*, the canonical target of irbesartan, was virtually undetectable in the first and second biopsies (0.04 and 0.01 RPKM, respectively).

To identify the cellular functions and pathways most impacted following treatment, we performed a gene set enrichment analysis on the differentially expressed genes (Methods). The genes more abundantly expressed in the second biopsy were significantly enriched for ontologies related to immune system processes (Figure 3C, Appendix A), including the regulation of cytokine production, leukocyte migration and the adaptive immune response. The genes more abundantly expressed in the first biopsy were enriched for processes including the AP-1 transcriptional network, negative regulation of cell proliferation and regulation of transcription in response to stress (Figure 3C).

### 2.4. Immune Infiltration Increases following Treatment with Irbesartan

The findings with gene expression pointed to differences in immune-related processes between the biopsies; therefore, we sought to evaluate the tumour immune microenvironment across both samples. First, we inferred the presence of immune cells using CIBERSORT (Figure 4A, Appendix A, Methods). The majority (68%, 15/22) of the immune cell types inferred with CIBERSORT were predicted to be present at a higher level in the second biopsy with 45% (10/22) having a fold-change of two or more. This is consistent with the gene ontology results and increased expression of T cell marker genes, such as CD8A (2.1-fold), CD8B (2.6-fold) and GZMA (3.4-fold).

To confirm the expression-based immune cell predictions from CIBERSORT [17] and examine the spatial profiles of the immune cells, we performed multiplex immunohistochemistry (m-IHC, see Methods) on the first and second biopsies as well as a sample from the initial diagnosis. The panels were designed to identify cytotoxic T cells (CD8+ GZMB+ CD3+), myeloid-derived suppressor cells (MDSCs) (CD11b+ CD33+ HLA-DR-negative) and B cells (CD20+ CD79a+, Figure 4B, Appendix A), as well as a panel with CD3 and FOS to confirm the high expression of *FOS* in the tumour cells.

Consistent with the results from our RNA-seq analyses, m-IHC revealed a significantly higher abundance of T cells (CD3+, mean 4-fold epithelial [*p* = 0.015], 19-fold stromal [*p* = 2.2 × 10^−6^]), particularly cytotoxic T cells (mean 23-fold epithelial [*p* = 2.12 × 10^−5^], 6-fold stromal [*p* = 6.62 × 10^−5^]), in the second biopsy compared to the first (Figure 4B,C) in both stromal and epithelial compartments. The level of T cells in the second biopsy was also higher (cytotoxic T cells: 40-fold epithelial [*p* = 1.72 × 10^−5^], 1.2-fold stromal [*p* = 0.013]) than that observed in an earlier diagnostic sample (Figure 4C), which is consistent with the notion of an immune response activated by treatment with irbesartan. B cells and MDSCs were also observed to be higher in the second biopsy compared to the first but only in the surrounding stromal compartment (B cells, mean −2.1-fold [*p* = 0.53] epithelial, 4.6-fold [*p* = 6.32 × 10^−4^] stromal; MDSCs, −2.9-fold [*p* = 0.71] epithelial, 2.4-fold [*p* = 0.046] stromal, Appendix A).

### 2.5. Second Biopsy Shows an Increased Diversity of T Cell Receptors

To further investigate the changes in the immune microenvironment that correlated with irbesartan treatment, we leveraged the RNA-seq data to explore the repertoire of T cell receptors (TCRs) present in each sample (Methods, Appendix A). These highly variable heterodimeric receptors drive immune responses through antigen recognition, and the expansion of a particular TCR clone may be associated with an immune response against a particular antigen. Twenty-eight β-chain clones were detected in the first biopsy. Consistent with the gene expression and m-IHC results, a higher number of clones (*n* = 182, 6.5-fold increase) was found in the second biopsy. In addition to an increase in clone count, the second biopsy also had a more diverse repertoire of β-clones (Shannon diversity 4.6 biopsy 2 vs. 2.9 biopsy 1, Figure 4D). Nine β-chain clones were shared between the two biopsies, the majority of which exhibited a lower frequency in the second biopsy (Figure 4E). One clone (#7) with the CDR3 sequence CASSSRTGELFF was detected at an increased frequency (2-fold increase) following treatment with irbesartan, and one remained at a constant frequency (1.1-fold increase, #8 CSAPDLPKSTDTQYF). It is currently unclear which antigen may have promoted the expansion of clone 7 and whether it may have contributed to the therapeutic response. This ß-chain clone was previously reported in response to a HIV-1 Gag epitope [18]. However, there was no evidence of HIV-1 in this patient, nor do any of this patient’s HLA alleles correspond to those associated with the Gag epitope.

### 2.6. Evidence of Immune Exhaustion in the Second Biopsy

We further explored the immune microenvironment for evidence of immune tolerance or T cell exhaustion. The percentage of tumour (epithelial) cells expressing PDL-1 was significantly increased in the second biopsy (Figure 5A,B, 13-fold, *p* = 0.001) as was the percentage of PD-L1+ stromal cells (10-fold, *p* = 1.12 × 10^−5^). PD1+ CD8+ cells were also more abundant in the second biopsy compared to the first (mean 16-fold epithelial [*p* = 0.0071], mean 78% of CD8+ cells were PD+ vs. 45%; 13-fold stromal [*p* = 1.72 × 10^−4^], mean 77% vs. 65%), indicating that cytotoxic cells may have been inhibited or exhausted at the time of relapse. These findings were corroborated by the TIDE expression signature [19], indicating a higher immune dysfunction in the second biopsy (−0.88 biopsy 1 vs. −1.15 biopsy 2).

## 3. Discussion

Comparative analysis of genome and transcriptome sequences from the first and second biopsies was consistent with immune activation during treatment with irbesartan. This observation correlated with the rapid and durable clinical response observed in the patient after five weeks [9]. We noted that both biopsies shared dominant mutation clusters at similar frequencies, and acquisition of new copy number alterations did not correlate with treatment. We thus propose that irbesartan treatment may not have been acting to select against clones carrying specific driver alterations, as we did not observe a gain or loss of driver events, but instead may have driven a broad anti-tumour effect sufficient to produce a sustained reduction in tumour burden. This patient experienced severe side effects to the standard chemotherapies, yet no side effects were noted during the irbesartan treatment except for low blood pressure, which was an expected consequence of this antihypertensive agent.

It is of note that the first biopsy exhibited markers that are associated with a clinical benefit to immunotherapies, including mismatch repair deficiency, high mutation burden and high predicted CD8+ T cell scores [5,20,21]. Given the increased immune expression signatures and infiltration observed in the second biopsy, we hypothesise that irbesartan may have acted through an immunological mechanism involving enhancement of the T cell and possibly B cell responses [22], which is an interesting topic for future mechanistic studies. Interestingly, the patient has remained on irbesartan since the treatment was initiated (~4 years on irbesartan), as other metastatic lesions (i.e., apart from the spinal lesion studied here) remained under control. Due to the spinal disease, nivolumab with four cycles of induction ipilimumab were given alongside irbesartan for approximately five months; however, this treatment was discontinued due to dermatological toxicities. The disease in the spine responded to the ICI–irbesartan treatment and continues to have a durable clinical response presently.

Immune checkpoint inhibitors are approved for use in MMR-deficient colorectal cancers as well as more recently for use in treating tumours with high TMB, agnostic of histology [23]. Good response rates in these selected groups are noted (e.g., an objective response rate of 40% in dMMR CRCs with pembrolizumab [20]). However, the cost of these medications can be prohibitive (mean $144,000 per year [24]). ARBs and angiotensin converting enzyme inhibitors (ACEIs) are substantially less expensive and, in combination with beta blockers (BBs), are associated with improved survival (OR = 0.27, *p* = 0.03) and fewer cancer-related hospitalizations (hospitalization ratio per year survived from ~3.5 to 1, *p* = 0.006) in patients with CRC [25]. The literature surrounding irbesartan and immune modulation is not well established and often conflicting; off-target activation of *PPARγ* has been described, which may activate inflammatory pathways [26,27]. In human umbilical vein endothelial cells, it has been reported that irbesartan inhibited TNF-α signalling of cell adhesion molecules potentially slowed the progression of inflammatory diseases [28]. Additionally, there is a case report of a hypersensitivity reaction to irbesartan, which may point to a drug interaction with the MHC and overactivation of the immune system [29]. Expression of AP-1-related genes was lower in the second biopsy as was protein expression of FOS, which suggests that some of the effect of irbesartan could be attributable to downregulation of the AP-1 complex.

Although this is a single case study, and there may be other potential mechanisms for the response observed, we present evidence linking involvement of the immune system with a prolonged clinical benefit for a patient with colorectal cancer who was treated solely with irbesartan. Further studies are warranted to fully understand the mechanism behind the response, and how the use of ARBs may be applicable to other patients with CRC.

## 4. Materials and Methods

### 4.1. Patient Details and Consent

Informed written consent was obtained from the patient for whole genome and transcriptome sequencing as part of the Personalized OncoGenomics (POG) research program and was approved by the UBC (University of British Columbia) BC Cancer Research Ethics Board. The patient’s age at initial diagnosis was 67 years, and the sex of the patient is female (XX). Normal control DNA was obtained from the patient’s peripheral blood to provide a comparator for somatic mutation calling. The original diagnostic biopsy was an FFPE sample taken from the retroperitoneum. The POG biopsy samples were taken from the recurrent lesion in the L3 spinous process. Whole genome and transcriptome analyses were performed on the two POG biopsies. The POG biopsy samples had comparable tumour content (63% and 55% bioinformatics estimated).

### 4.2. Whole Genome and Transcriptome Sequencing

Whole genomes and poly-adenylated RNA were sequenced as previously described [7,8,9]. Tumour whole genomes were sequenced to a coverage of 86 X and 93 X (biopsies 1 and 2, respectively) and peripheral blood to 43 X. Transcriptomes were sequenced to 159 million and 191 million reads for biopsies 1 and 2, respectively. The RNA-seq reads were converted to RPKMs for analysis as previously described [9].

### 4.3. Mutation Calling and Copy Number Alterations

Whole genome sequencing reads were aligned to the reference human genome (hg19) as previously described [8]. Copy number alterations and regions with loss of heterozygosity were detected using CNAseq [30] and APOLLOH [7]. Single nucleotide variants were detected using a joint variant calling approach with SAMtools [31], MutationSeq [32] and Strelka [33] and small insertions and deletions using Strelka. Mutations were correlated against a catalogue of known mutation signatures (COSMIC [34]) using an NNLS approach. TMB was calculated using TMBur [35].

### 4.4. Transcriptome Analysis

A database of exon junction sequences was used to align the RNA-Seq reads to hg19 using Jaguar [36]. In-house processing was then used to determine the gene and exon read counts, normalised to reads per kilobase per million mapped reads (RPKM).

### 4.5. Clonal Evolution

PyClone [16] was used to identify mutation clones, consisting of mutations with similar shifts in frequency in data from both biopsies. As PyClone was designed for deeply redundant sequencing data (e.g., ~1000×), 100,000 iterations were performed (first 10,000 discarded), and a binomial density model was used to analyse the whole genome sequencing. Bioinformatics-estimated tumour content (as described in Pender et al. [5]) was also provided as an input for each of the biopsies to obtain the predicted cellular prevalence of each mutation.

### 4.6. Gene Set Enrichment Analysis

Genes with a change in transcript abundance between the two biopsies of at least 2-fold in either direction (*n* = 907 more abundant, *n* = 194 less abundant) were used for functional enrichment analyses using Metascape [37]. The functional pathways tested for enrichment included all GO (Gene Ontology) [38,39] terms. These terms are clustered by Metascape into their master groups, shown in Figure 3, which reduces redundancy between overlapping functional processes.

### 4.7. Immune Cell Deconvolution Predictions

The RNA-seq data were deconvoluted to infer the presence of immune cells using CIBERSORT as previously described [8]. The outputs for both biopsy samples had *p*-values < 0.05.

### 4.8. Immunohistochemistry Staining

Five different multicolour immunohistochemical panels were selected to analyse the tissue using 3 general staining schemes. Unless stated otherwise, all reagents were sourced from Biocare Medical (Pacheco, CA, USA). All slides were incubated overnight at 37 °C and then deparaffinised using xylene and graded alcohols. Then, the slides were subjected to antigen retrieval in a Biocare Decloaking chamber at 110 °C for 15 min in Diva decloaking solution and loaded onto a Biocare Intellipath FLX autostainer. Endogenous peroxidise activity was blocked with peroxidase-1 for 5 min followed by blocking of non-specific binding with background sniper for 10 min. All antibodies were diluted in Biocare’s Da Vinci green diluent. The first staining scheme involved one antibody in the first round of staining that was detected using IP Ferangi Blue Chromogen followed by a denaturation step with SDS-glycine pH 2.0 at 50 °C for 45 min (Pirici et al.); then, a second round of staining was performed with the remaining 2 antibodies cocktailed and detected with IP Warp Red Chromogen and Hi Def Yellow Chromogen (Enzo, Farmingdale, NY, USA). This scheme was used for the Granzyme B, CD8, CD3 and HLA-DR, CD33 and CD11b panels. Either Granzyme B (clone GrB-7, Thermo Scientific, Waltham, MA, USA) or HLA-DR (clone EPR3692, Abcam, Toronto, ON, Canada) was added for 30 min followed by a 30 min incubation with Mach2 Mouse-AP polymer or Mach2 Rabbit-AP polymer, respectively. In the second round of staining, a cocktail of either CD8 (clone C8/144B, Cell Marque, Rocklin, CA, USA) and CD3 (clone SP7, Abcam) or CD33 (clone 6C5/2, Abcam) and CD11b (clone EPR1344, Abcam) was added to the slide for 30 min followed by Mach 2 Double Stain #2 polymer or Mach 2 Double Stain #1 polymer for 30 min. Following the chromogen step, the slides were counterstained with Cat Hematoxylin at a 1/5 dilution and then washed and air dried prior to cover slipping with Ecomount. The second staining scheme was a double simultaneous stain using IP Warp Red and IP DAB chromogens. A cocktail of either CD20 (clone L26, Biocare) and CD79a (clone SP18, Abcam) or CD3 (clone SP7, Abcam) and FOS (clone 2H2, Abcam) was added to the slide for 30 min followed by Mach2 Double Stain #2, chromogens, hematoxylin and cover slipping. For the last panel, a cocktail of PDL1 (clone SP142, Abcam) and PD1 (clone NAT105, Cell Marque) was added to the slides followed by Mach2 Double Stain #1 and IP Ferangi blue (8 min) and IP DAB (5 min) chromogens and a denaturation step as described above. The second round of staining used CD8 (C8/144B, Cell Marque) followed by Mach 2 Mouse-AP polymer, IP Warp Red Chromogen (7 min), hematoxylin and cover slipping.

### 4.9. Immunohistochemistry Counts

All slides were then scanned using a Vectra automated imaging system (Perkin Elmer, Waltham, MA, USA) at 4×, and 10 fields were captured for each slide at 20×. inForm image analysis software was used to process the images (Perkin Elmer, Waltham, MA, USA), running tissue segmentation algorithms to distinguish the cells present in the tissue stroma (stromal compartment) from the cells found within the tumour tissue (epithelial compartment) and generating cell phenotype counts for each slide, defined as the combination of markers that each cell expresses. The counts were inspected visually to ensure the results were reliable. The tissue segmented regions (epithelial or stromal) for each slide were converted from the number of pixels to the area of tissue (mm^2^). The average cell counts for each phenotype were then normalised by the segmented area to obtain the cell density counts for each region, which were then compared across the biopsy samples. The normalised cell counts for each biopsy were plotted in R, and significance values between the first and second biopsies were calculated using a two-tailed Wilcoxon rank sum test.

### 4.10. HLA Typing

The patient’s HLA-I alleles were genotyped to their four-digit code using Optitype [40] with both the pre- and post-treatment RNA-Seq data.

### 4.11. T Cell Receptor Analysis

The landscape of infiltrating T cell receptors (TCRs) was analysed using MiXCR [41], VDJtools [42] and tcR [43]. The TCR sequences were identified from the transcriptome data for each biopsy using MiXCR and run according to the RNA-Seq workflow described in the package documentation. Briefly, the reads were aligned against the reference V, D, J and C genes, and full CDR3 regions were assembled and exported as clone sequences for TCR β chains. Contigs were built for the reads that only partially aligned to the CDR3 region, which were then used for another round of alignment. The TCR β clones were then exported to VDJtools for comparisons between the two biopsies and data visualisation, where non-functional receptor sequences were filtered out. VDJdb [44], a curated database of T cell receptor sequences, was used to obtain further information about clonotypes of interest detected in the patient samples. The Shannon diversity was calculated using the R package vegan v2.5.7.

## Figures and Tables

**Figure 1 ijms-24-05869-f001:**
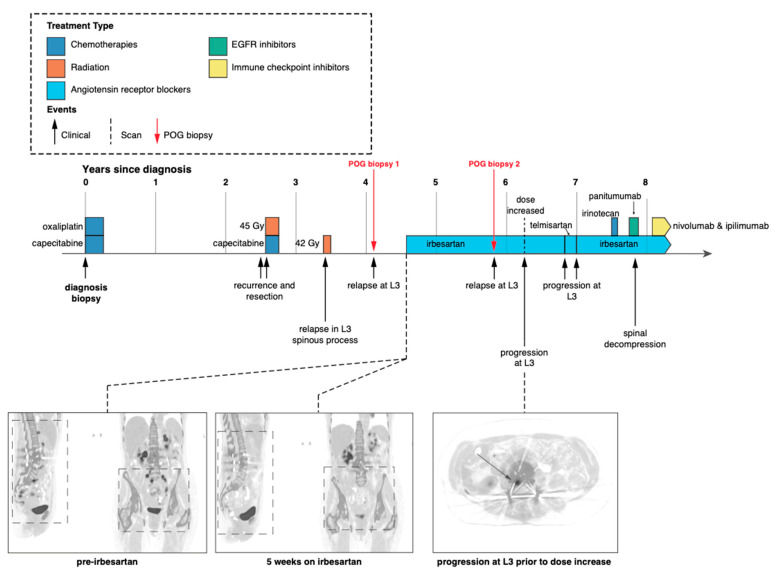
Patient clinical and treatment history. Timeline depicting patient treatment, follow-up and imaging since initial diagnosis [9]. Red arrows indicate biopsies for the POG program described in this study. Both POG biopsies were taken from the same location at the base of the spine.

**Figure 2 ijms-24-05869-f002:**
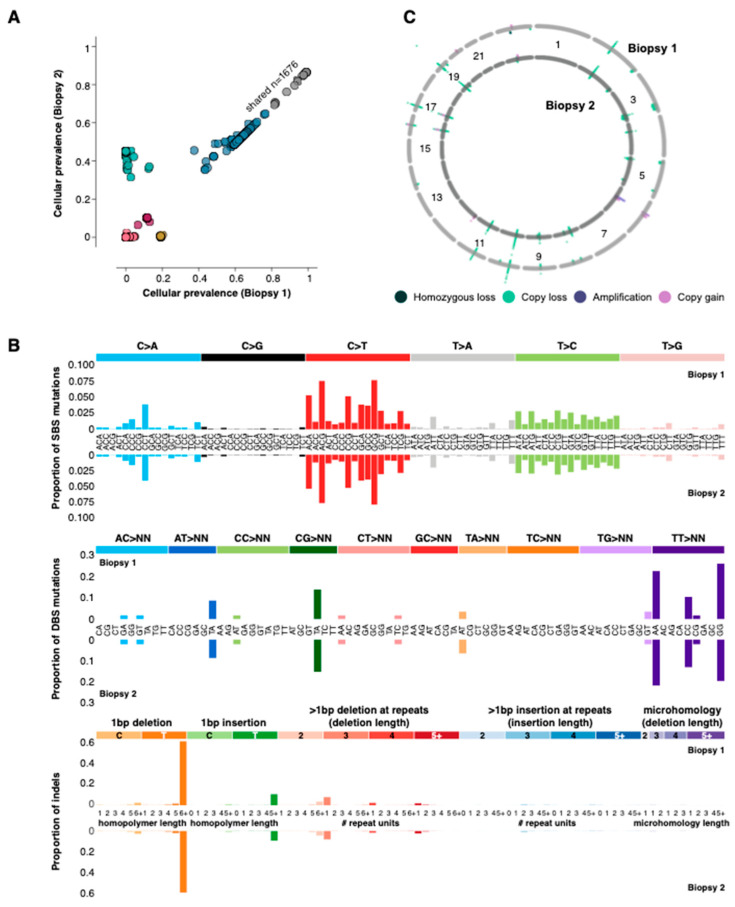
Genomic alterations. (**A**) Cellular prevalence [16] of single nucleotide variants identified in the two biopsies. The variants are coloured by the cluster to which they were assigned. (**B**) The proportion of single base, double base and indel mutations associated with specific base changes (mutation signatures). Biopsy 1 is on the top row for each mutation type, and biopsy 2 is on the bottom. (**C**) Copy number landscape of the two biopsies. The first biopsy is represented by the outer ring and the second biopsy by the inner ring.

**Figure 3 ijms-24-05869-f003:**
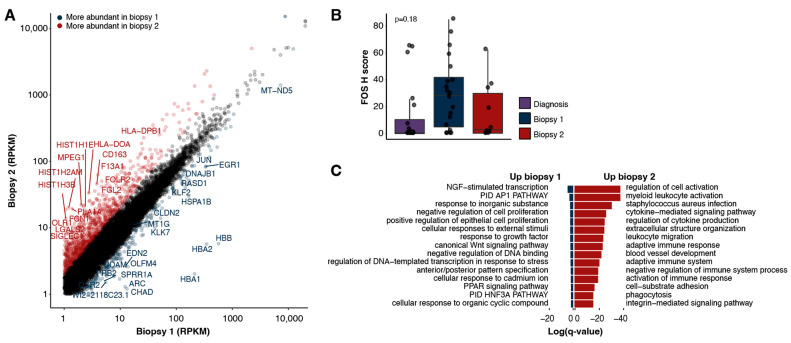
Expression landscapes. (**A**) Expression of protein-coding genes detected in the two biopsies (RPKMs). Genes identified as up-regulated (≥2-fold) in the second biopsy are indicated in red, and those up-regulated in the first biopsy are in blue (≥2-fold). (**B**) H score for FOS as measured with immunohistochemistry for biopsies one, two and the earlier diagnostic sample. The *p*-value is determined with a two-tailed Wilcoxon rank sum test between biopsy one and two. (**C**) Gene ontologies enriched in either the first (left) or second (right) biopsies.

**Figure 4 ijms-24-05869-f004:**
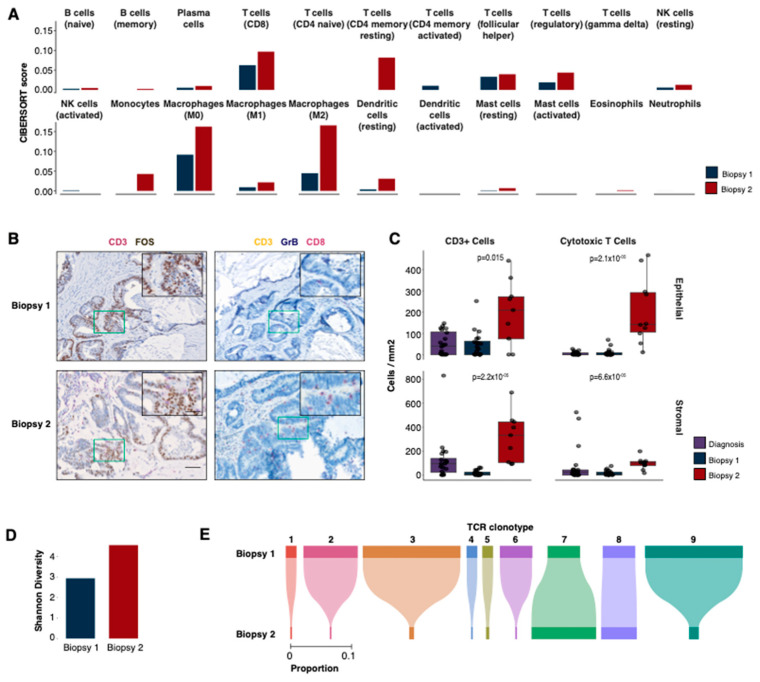
Immune profiles. (**A**) Immune deconvolution predictions for 22 immune cell types using CIBERSORT. (**B**) Immunohistochemistry (IHC) staining for two panels (CD3, FOS; CD3, GrB and CD8) in biopsies 1 (upper row) and 2 (bottom row). The colour stain for each protein is indicated in the label above each image. (**C**) IHC cell counts/mm^2^ for different cell types in biopsies 1 and 2 and the diagnostic sample. The cell counts are reported separately for epithelial (top) and stromal (bottom) compartments. The *p*-values displayed are two-tailed Wilcoxon rank sum tests between the first and second biopsies. (**D**) The Shannon diversity of TCR β-chain repertoires for both biopsies. (**E**) The proportions of shared β-chain clones between biopsies 1 and 2.

**Figure 5 ijms-24-05869-f005:**
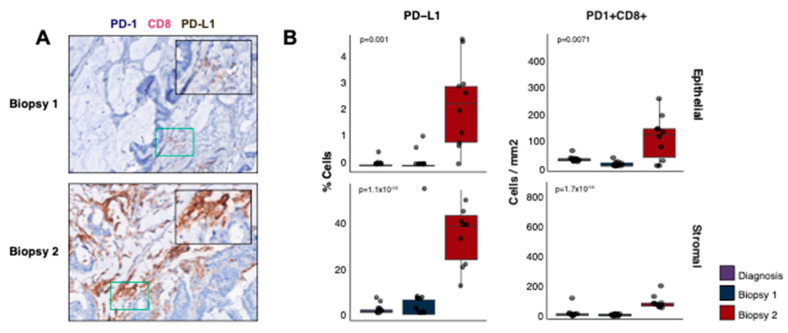
Immune exhaustion and resistance. (**A**) m-IHC staining for PD-1, CD8 and PD-L1 in biopsies one (top) and two (bottom). The colour stain for each protein is indicated in the label above the images. (**B**) The percentage of cells expressing PD-L1 (top row) and IHC cell counts/mm^2^ for PD-1+ CD8+ cells (bottom row) in biopsies one, two and the diagnostic sample. The cell counts are split into epithelial (left) and stromal (right) compartments. The *p*-values displayed are two-tailed Wilcoxon rank sum tests between the first and second biopsies.

## Data Availability

Genomic and transcriptome sequence datasets for the POG program are available at (EGA, http://www.ebi.ac.uk/ega/ (accessed on 18 March 2023)) as part of the study EGAS00001001159 [8], patient ID 23674. Data is also available for view at https://www.personalizedoncogenomics.org/cbioportal/ (accessed on 18 March 2023). TCR sequences identified in both biopsies are available in Appendix A. All other data are available from the corresponding author on reasonable request.

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
