# Peer review of "Immune Activation following Irbesartan Treatment in a Colorectal Cancer Patient: A Case Study"

_ijms, 2023, doi:10.3390/ijms24065869_

Round 1
Reviewer 1 Report
This is a single case, and the evidence presented, despite suggesting the conclusions that the authors point to, cannot rule out other types of interactions. It should be treated with caution as a clinical case description and not as a research article per se.
Author Response
We thank the reviewer for their time in critiquing our study. We agree that there may be alternative interactions occurring as it is a single case study and analysis is limited. In the discussion, we mentioned PPAR? as a potential alternative mechanism, as well as interaction with the MHC in a hypersensitivity context. To address the reviewer’s comment, we have tried to expand on this and present these ideas more directly as alternative mechanisms, as well as introduce other literature.
Additionally, we edited the last paragraph of the discussion to try and make this point more directly:
“Although this is a single case study, and there may be other potential mechanisms for the response observed, we present evidence linking involvement of the immune system with a prolonged clinical benefit for a colorectal cancer patient treated solely with irbesartan. Further studies are warranted to fully understand the mechanism behind the response, and how use of ARBs may be applicable to other CRC patients.”
Reviewer 2 Report
Reviewer’s Comments:
The manuscript “Evidence of immune activation in a colorectal cancer patient following a clinical response to irbesartan, an angiotensin receptor blocker” is a very interesting work. In this work, Colorectal cancers are one of the most prevalent tumour types worldwide and, despite the emergence of targeted and biologic therapies, have among the highest mortality rates. The Personalized OncoGenomics (POG) program at BC Cancer performs whole genome and transcriptome analysis (WGTA) to identify specific alterations in an individual’s cancer that may be most effectively targeted. Informed by WGTA, a patient with advanced mismatch repair deficient colorectal cancer was treated with the antihypertensive drug, irbesartan, and experienced a profound and durable response. We describe the subsequent relapse of this patient and potential mechanisms of response, using WGTA and multiplex immunohistochemistry (m-IHC) profiling of biopsies before and after treatment from the same metastatic site of the L3 spine. While I believe this topic is of great interest to our readers, I think it needs major revision before it is ready for publication. So, I recommend this manuscript for publication with major revisions.
1. In this manuscript, the authors did not explain the importance of the immune activation the introduction part. The authors should explain the importance of immune activation.
2) Title: The title of the manuscript is not impressive. It should be modified or rewritten it.
3) Correct the following statement “Expression profiling and mIHC revealed increased expression of immune related genes, activation of immune signalling pathways and increased presence of infiltrating immune cells, particularly CD8+ T cells, in the relapsed tumour. Our evidence indicates that the observed anti-tumour response to irbesartan may have been due to an activated immune response”.
4) Keywords: The immune activation is missing in the keywords. So, modify the keywords.
5) Introduction part is not impressive. The references cited are very old. So, Improve it with some latest literature like 10.1016/j.molstruc.2021.131145, 10.1016/j.jallcom.2021.159013
6) The authors should explain the following statement with recent references, “To confirm the expression-based immune cell predictions and examine spatial profiles of immune cells, we performed multiplex immunohistochemistry (m-IHC) on the first and second biopsies, as well as a sample from the initial diagnosis”.
7) Add space between magnitude and unit. For example, in synthesis “21.96g” should be 21.96 g. Make the corrections throughout the manuscript regarding values and units.
8) The author should provide reason about this statement “We thus propose that irbesartan treatment may not have been acting to select against clones carrying specific driver alterations, but instead may have driven a broad anti-tumour effect sufficient to produce a sustained reduction in tumour burden”.
9. Comparison of the present results with other similar findings in the literature should be discussed in more detail. This is necessary in order to place this work together with other work in the field and to give more credibility to the present results.
10) Conclusion part is very long. Make it brief and improve by adding the results of your studies.
11) There are many grammatic mistakes. Improve the English grammar of the manuscript.
Author Response
We thank the reviewer for their time in critiquing our study and providing suggestions for improvement. We have responded to individual comments below in italics.
- In this manuscript, the authors did not explain the importance of the immune activation in the introduction part. The authors should explain the importance of immune activation.
We thank the reviewer for highlighting that this was missing from the introduction. We have added a statement in the introduction linking immune presence at the tumour site with ICI response.
2) Title: The title of the manuscript is not impressive. It should be modified or rewritten it.
We have modified the title to “Immune activation following irbesartan treatment in a colorectal cancer patient; a case study”. We agree that the title should be impressive, but also want to keep it clear that it is a case study of a single patient.
3) Correct the following statement “Expression profiling and mIHC revealed increased expression of immune related genes, activation of immune signalling pathways and increased presence of infiltrating immune cells, particularly CD8+ T cells, in the relapsed tumour. Our evidence indicates that the observed anti-tumour response to irbesartan may have been due to an activated immune response”.
We have shortened the statements above to: “Analyses revealed an increase in immune signalling, and infiltrating immune cells, particularly CD8+ T cells, in the relapsed tumour. These results indicate that the observed anti-tumour response to irbesartan may have been due to an activated immune response.”
4) Keywords: The immune activation is missing in the keywords. So, modify the keywords.
We have modified the keywords to include immune activation.
5) Introduction part is not impressive. The references cited are very old. So, Improve it with some latest literature like 10.1016/j.molstruc.2021.131145, 10.1016/j.jallcom.2021.159013
We thank the reviewer for highlighting that the introduction can be improved with some updated literature. We have included new references describing the association between ICI response and immune presence in the introduction, as per comment #1, and have updated literature supporting existing points in the introduction (e.g. Quintanilha et al., 2023).
As some of the introduction is describing the initial presentation and response of the patient, there is inclusion of text and references linked to that study and the POG program which are old (2012-2016), but important for context. We considered inclusion of the articles suggested by the reviewer, however, we believe the topics are not directly relevant to the current study (design of a carbonic anhydrase inhibitor, and bioactive glass nanoparticles).
We have also added some more literature in the discussion as per comment #9.
6) The authors should explain the following statement with recent references, “To confirm the expression-based immune cell predictions and examine spatial profiles of immune cells, we performed multiplex immunohistochemistry (m-IHC) on the first and second biopsies, as well as a sample from the initial diagnosis”.
We have included a reference for CIBERSORT when mentioning the immune cell predictions (line 167), as well as a referral to the Methods section of the paper for details on the m-IHC methodology. There are no recent references to add to this statement as it is introducing the next result.
7) Add space between magnitude and unit. For example, in synthesis “21.96g” should be 21.96 g. Make the corrections throughout the manuscript regarding values and units.
We have updated all instances of values and SI units to include spaces.
8) The author should provide reason about this statement “We thus propose that irbesartan treatment may not have been acting to select against clones carrying specific driver alterations, but instead may have driven a broad anti-tumour effect sufficient to produce a sustained reduction in tumour burden”.
We have expanded the sentence above in an attempt to clarify the rationale for the statement. With the preceding sentence this section now reads: “We noted that both biopsies shared dominant mutation clusters at similar frequencies and that acquisition of new copy number alterations did not correlate with treatment. We thus propose that irbesartan may not have been acting to select against clones carrying specific driver alterations, as we did not observe a gain or loss of driver events, but instead may have driven a broad anti-tumour effect sufficient to produce a sustained reduction in tumour burden.”
- Comparison of the present results with other similar findings in the literature should be discussed in more detail. This is necessary in order to place this work together with other work in the field and to give more credibility to the present results.
As this is a descriptive case study, there is somewhat limited literature surrounding the topic of angiotensin receptor blockers and immune activation, and/or cancer therapy. However, we have tried to include other relevant literature in the discussion that may aid with context.
10) Conclusion part is very long. Make it brief and improve by adding the results of your studies.
To address comment #9, we have included some additional literature in the discussion, but also tried to shorten in other parts, for example the last paragraph.
11) There are many grammatic mistakes. Improve the English grammar of the manuscript.
We thank the reviewer for thoroughly reviewing the grammar in the manuscript. We have addressed the following statements to improve readability and grammar, and will be happy to make any further edits as per the editorial team’s requests.
Line 326, “Genes that were more or less abundant in the second biopsy” to “Genes with a change in transcript abundance between the two biopsies”.
Line 346, “Outputs for both first and second biopsy samples” to “Outputs for both biopsy samples”.